# Spot the Difference: Detection of Topological Changes via Geometric Alignment

**Steffen Czolbe**
Department of Computer Science
University of Copenhagen
`per.sc@di.ku.dk`

**Aasa Feragen**
DTU Compute
Technical University of Denmark
`afhar@dtu.dk`

**Oswin Krause**
Department of Computer Science
University of Copenhagen
`oswin.krause@di.ku.dk`

## Abstract

Geometric alignment appears in a variety of applications, ranging from domain adaptation, optimal transport, and normalizing flows in machine learning; optical flow and learned augmentation in computer vision and deformable registration within biomedical imaging. A recurring challenge is the alignment of domains whose topology is not the same; a problem that is routinely ignored, potentially introducing bias in downstream analysis. As a first step towards solving such alignment problems, we propose an unsupervised algorithm for the detection of changes in image topology. The model is based on a conditional variational auto-encoder and detects topological changes between two images during the registration step. We account for both topological changes in the image under spatial variation and unexpected transformations. Our approach is validated on two tasks and datasets: detection of topological changes in microscopy images of cells, and unsupervised anomaly detection brain imaging.

## 1 Introduction

Geometric alignment is a fundamental component of widely different algorithms, ranging from domain adaptation [7], optimal transport [40] and normalizing flows [35, 42] in machine learning; optical flow [21, 51] and learned augmentation [20] in computer vision, and deformable registration within biomedical imaging [5, 15, 19, 39, 53]. A recurring challenge is the alignment of domains whose topology is not the same. When the objects to be aligned are probability distributions [35], this appears when distributions have different numbers of modes whose support is separated into separate connected components. When the objects to be aligned are scenes or natural images, the problem occurs with occlusion or temporal changes [51]. In biomedical image registration, the problem is very common and happens when the studied anatomy differs from "standard" anatomy [36]. Despite being extremely common, this problem is routinely ignored or accepted as inevitable, potentially introducing bias in downstream analysis.

We study two cases from biomedical image registration. One is the alignment of image slices to reconstruct a 3d volume, where changes in topology between slices introduce challenges in post-processing (Figure 1). The other is the registration of brain MRI scans, where tumors give common examples of anatomies that are topologically different from healthy brains. In deformable image registration, a "moving image" is mapped via a nonlinear transformation to make it as similar as possible to a "target" image, enabling matching local features or transferring information from one

35th Conference on Neural Information Processing Systems (NeurIPS 2021).

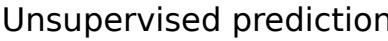

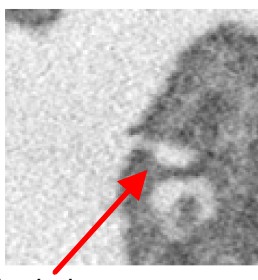
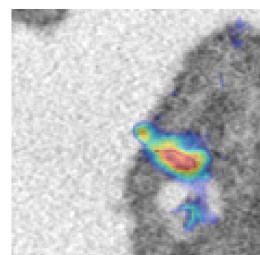

Topological change

Figure 1: Left: Example of topological changes between two adjacent slices of human blood cells imaged via serial block-face scanning electron microscopy [41]. We aim to detect the change of topology caused by an emerging organelle within the cell (highlighted by the red arrow) while accounting for non-linear deformations of the image introduced by natural shape changes between slices. Right: Heatmap of the likelihood of topological changes predicted by our unsupervised model.

image to another. It is common to numerically stabilize the estimation of the transformation by constraining the predicted transformation to be diffeomorphic, that is, bijective and continuously differentiable in both directions. In particular, diffeomorphic transformations are homeomorphic, or topology-preserving, which implies that a common topology is assumed across all images [13, 15]. This topology is often provided by a common template image $\mathbf{I}_{\text{template}}$, from which all other images are obtained via the transformation $\Phi$ from the group of diffeomorphisms $\mathcal{G}$. Under this common topology assumption, the set of all images is given by

$$\mathcal{I} = \{\mathbf{I}_{\text{template}} \circ \Phi | \Phi \in \mathcal{G}\} \ .$$

Topological differences in biomedical images can be caused by a variety of processes. For instance, image slices obtained from a volume do not all contain the same elements. Tumor growth or the removal of surgical tissue can alter the topology of an image. Various processes can lead to the replacement or deformation of organic tissue, which cannot be mapped to the original image. We choose to model these topological differences as the inability to obtain one image from the other via a homeomorphic transformation of the image domain. Since, within image registration, transformations are assumed to be continuously differentiable, we are effectively modelling topological differences between pairs of images via the failures of diffeomorphic image registration in aligning them.

As most registration algorithms align images based on intensity, e.g. minimizing mean squared error (MSE), these tissue changes make it difficult to map images correctly. The strong local deformations required to deal with the non-diffeomorphic part of the image inevitably also deform the surrounding area, leading to distorted transformation fields in topologically matching parts of the image [36]. These transformation fields adversely affect downstream tasks, for example indicating false size changes in adjacent regions.

**Previous work on aligning topologically inconsistent domains.** Attempting to relax the same-image assumption induced by fully diffeomorphic transformations is not new. In the context of organs sliding against each other, several approaches exist, most of which rely on pre-annotating the sliding boundary using organ segmentation [6, 10, 22, 37, 43, 46], with a few extensions to un-annotated images [38, 45].

When topological holes are created or removed in the domain, for example through tumors, pathologies, or surgical resections, the loss function used for registration can be locally weighted or masked [26, 29, 30], or an artificial insection can be grown to correct anatomies [36]. These approaches rely on annotation of the topological differences, which have to be provided manually or by segmentation. An exception is given by Li and Wyatt [30], which detects changes in topology from the difference between the aligned images. This depends crucially on the ability to find a good diffeomorphic registration *outside* the anomaly, which is difficult all the while the applied transformation is still diffeomorphic.

An alternative approach to registering topologically inconsistent images is to inpaint the difference in the source images to obtain a topologically consistent quasi-normal image. Then standard registration methods can be used on the altered images. Quasi-normal images can be obtained through low-rank and sparse matrix decomposition [32, 33], principle component analysis [16, 18], denoising VAEs [52], or learning of a blended representation [17]. Registration with the quasi-normal approach retains the diffeomorphic properties of the transformation but does not register the topologically inconsistent areas of the images.

**Our contribution.** We propose an unsupervised algorithm for the detection of changes in image topology. To this end, we train a conditional variational autoencoder for predicting image-to-image alignment, obtaining a per-target-pixel probability of being obtained from the moving image via diffeomorphic transformation. We combine a semantic loss function trained to extract contextual information [8], with a learnable prior of transformations [9], allowing us to incorporate both the reconstruction error, as well as knowledge about the expected transformation strength.

We test the validity of our approach on a novel dataset of cell slices with annotated topological changes and on the proxy task of unsupervised brain-tumor detection. We also validate our approach by investigating a spatial "topological inconsistency likelihood", and showing that this likelihood is higher in regions where topological inconsistencies are known to be common. Our model is able to detect topological inconsistencies with a purely registration-driven framework, and thus provides the first step towards an end-to-end registration model for images with topological discrepancies. The implementation is available at github.com/SteffenCzolbe/TopologicalChangeDetection.

## 2 Background

### 2.1 Notation of images and transformations

We view an image $\mathbf{I}$ interchangeably as two different structures. First, it is a continuous function $\mathbf{I} : \Omega_{\mathbf{I}} \to \mathbb{R}^C$, where $\Omega_{\mathbf{I}} = [0,1]^D$ is the domain of the image, and $C$ the number of channels. This function can be approximated by a grid of $n$ pixels with positions $x_k \in \Omega_{\mathbf{I}}$ leading to the image representation $\mathbf{I}_k^{(c)}$, where c is an index over the channels and $\mathbf{I}_k = (\mathbf{I}_k^{(1)}, \dots, \mathbf{I}_k^{(C)})^T = \mathbf{I}(x_k)$. Second, this pixel grid is accompanied by a graph structure that encodes the neighbourhood of each pixel. In this view, the set of neighbours of a pixel with index $k$ (for example the 4-neighbourhood of a pixel on the image grid) is referred to as $N(k)$ and $|N(k)|$ is the number of neighbours. The neighborhoods of a pixel gives rise to a graph which can be described via the graph laplacian $\Lambda \in \mathbb{R}^{n \times n}$ with $\Lambda_{k,k} = |N(k)|$ and $\Lambda_{k,k'} = -1$ when pixel $k' \in N(k)$, and zero otherwise.

Applying a spatial transformation $\Phi : \mathbb{R}^D \to \mathbb{R}^D$ to an image is written as $\mathbf{J} = \mathbf{I} \circ \Phi$, which can be seen as its own image with domain $\Omega_{\mathbf{J}} = [0,1]^D$ with pixel coordinates $y_k \in \Omega_{\mathbf{J}}$ and $\mathbf{J}_k = \mathbf{I}(\Phi(y_k))$. The transformation $\Phi$ can be seen as a vector field on the image domain which assigns each pixel in $\mathbf{J}$ a position on $\mathbf{I}$ and thus it can be parameterized as a pixel grid $\Phi_k^{(d)}$, $d = 1, \dots, D$ at the pixel coordinates of $\mathbf{J}$ using $\Phi(y_k) = y_k + \Phi_k$. To make this choice of coordinate system clear, we will refer to a transformation that moves a pixel position from the domain $\Omega_{\mathbf{J}}$ to the corresponding pixel in domain $\Omega_{\mathbf{I}}$ as $\Phi_{\mathbf{J} \to \mathbf{I}}$, whenever it is not clear from the context. If $\Phi$ is a diffeomorphism, it can alternatively be parameterized by a vector field $V$ on the tangent space around the identity, where the mapping between the tangent space and the transformation is given by $\Phi = \exp(V)$, which amounts to integration over the vector field [2].

### 2.2 Variational registration framework

It is possible to phrase the problem of fitting a registration model in terms of variational inference, using an approach similar to conditional variational autoencoders [47]. Here, we summarize the approach taken by [9, 31]. For a $D$-dimensional image pair $(\mathbf{I}, \mathbf{J})$, we assume that $\mathbf{J}$ is generated from $\mathbf{I}$ by drawing a transformation $\Phi$ from a prior distribution $p(\Phi|\mathbf{I})$, apply it to $\mathbf{I}$ and then add pixel-wise noise:

$$p(\mathbf{J}|\mathbf{I}) = \int p_{\text{noise}}(\mathbf{J}|\mathbf{I} \circ \Phi)p(\Phi|\mathbf{I}) \, d\Phi$$

This includes the common topology assumption implicitly via $p(\Phi|\mathbf{I})$, which is typically chosen to produce invertible transformations depending only on the topology of $\mathbf{I}$, as well as the noise

model which does not assume systematic changes between $\mathbf{J}$ and $\mathbf{I}$. This model can be learned using variational inference using a proposal distribution $q(\Phi|\mathbf{I},\mathbf{J})$ with evidence lower bound (ELBO)

$$\log p(\mathbf{J}|\mathbf{I}) \geq E_{q(\Phi|\mathbf{I},\mathbf{J})}\left[\log p_{\text{noise}}(\mathbf{J}|\mathbf{I}\circ\Phi)\right] - KL(q(\Phi|\mathbf{I},\mathbf{J})\|p(\Phi|\mathbf{I})) \ . \tag{1}$$

In contrast to variational autoencoders, the decoder is given by the known application of $\Phi$ to $\mathbf{I}$. Thus, the degrees of freedom in this model are in the choice of the encoder, prior, and the noise distribution. Dalca et al. [9] proposed to parameterize $\Phi$ as a vector field $V_k^{(d)}$ on the tangent space, which turns application of $\Phi = \exp(V)$ into sampling an image with a spatial transformer module [24]. As a prior for this parameterization, they chose a prior independent of $\mathbf{I}$

$$p(\Phi) = \prod_{d=1}^{D} \mathcal{N}\left(V^{(d)} \mid 0, \Lambda^{-1}\right) \ ,$$

where we used the implicit identification of $\Phi$ and $V$ and the precision matrix $\Lambda$ is chosen as the Graph Laplacian over the neighbourhood graph (see notation). Using an encoder that for each pixel proposes $q(V_k^{(d)}|\mathbf{I},\mathbf{J}) = \mathcal{N}(\mu_k^{(d)}, v_k^{(d)})$, the KL divergence is derived as

$$\text{KL}\left(q(\Phi|\mathbf{I},\mathbf{J})\|p(\Phi|\mathbf{I})\right) = \frac{1}{2}\sum_{d=1}^{D}\sum_{k=1}^{n} -\log v_k^{(d)} + |N(k)|v_k^{(d)} + \sum_{l\in N(k)}\left(\mu_k^{(d)} - \mu_l^{(d)}\right)^2 + \text{const} \ . \tag{2}$$

It is worth noting that this equation is invariant under translations of $\mu$. This invariance manifests in rank-deficiency of $\Lambda$ and as a result, const is infinite. Thus, sampling from the prior and bounding the objective is impossible. Still training with this term works in practice as images are usually pre-aligned with an affine transformation and thus translations are close to zero. We will present a slightly modified approach, rectifying the missing eigenvalue.

## 3 Detection of topological differences

The variational approach for learning the distribution of transformations introduced before optimizes an ELBO on $\log p(\mathbf{J}|\mathbf{I})$. This information is enough to detect images that contain topological differences under the assumption that these images will overall have a lower likelihood. However, in our application, we need not only to detect the existence but also the position of outliers in the image. For this, we have to ensure that $\log p(\mathbf{J}|\mathbf{I})$ can be decomposed into a likelihood for each pixel of the image. It is immediately obvious by inspection of the ELBO (1) together with the KL-Divergence (2), that the lower bound on $\log p(\mathbf{J}|\mathbf{I})$ can be decomposed into pixel-wise terms if $\log p_{\text{noise}}(\mathbf{J}|\mathbf{I}\circ\Phi)$ can be decomposed as such. To enforce this, we will introduce a general form of error function, which can be decomposed and includes the MSE as a special case. For this, we first map the images $\mathbf{I}$ and $\mathbf{J}$ to feature maps over the pixel positions $k$ via a mapping $f_k(\mathbf{I}) \in \mathbb{R}^F$ and define the loss as:

$$p_{\text{noise}}(\mathbf{J}|\mathbf{I}\circ\Phi) = \prod_{k=1}^{n} \mathcal{N}(f_k(\mathbf{J})|f_k(\mathbf{I})\circ\Phi, \Sigma_f) \ , \tag{3}$$

where $\Sigma_f \in \mathbb{R}^{F\times F}$ is a diagonal covariance matrix with variances learned during training.

The ability to decompose the likelihood is not enough for a *meaningful* metric, as we have to ensure that each term is calculated in the correct coordinate system. This depends on the parameterisation and regularisation of $\Phi$. In the approach by Dalca et al. [9] the parameterization $V$ of $\Phi$ is defined on the tangent space and consequently the prior is also on this space. Since the connection between $\Phi$ and $V$ is given by integration of the vector field, decomposing (2) for a single pixel $k$ will produce estimates based on the local differential of the transformation, but will not take the full path with starting and endpoints into account. Thus, correct cost assignments require integration of (2) over the computed path, which is expensive and suffers from severe integration inaccuracies. Instead, we will use an alternative approach, where we parameterize $\Phi$ directly as a vector field on the image domain. Transformations parameterized this way are not necessarily invertible anymore, yet smoothness is still encouraged by the prior.

**Learnable prior** Using this parameterization, we extend the approach by Dalca et al. [9] and introduce a parameterized prior on $\Phi_k$ that is learned simultaneously with the model:

$$p(\Phi) = \prod_{d=1}^{D} \mathcal{N}\left(\Phi^{(d)} \mid 0, \Lambda_{\alpha\beta}^{-1}\right), \ \ \Lambda_{\alpha\beta} = \alpha\Lambda + \frac{\beta}{n^2}\mathbb{1}\mathbb{1}^T \tag{4}$$

The expected variations and translations between transformation vectors are governed by $\alpha$ and $\beta$. Unlike most works in image registration, we do not treat these as tuneable hyperparameters, but instead view them as unknowns to be fitted to the data during training similar to [28, 49]. For efficient learning, we use an estimate for the optimal values for $\alpha, \beta$ over a batch of samples during training, and use a running average at test time. A detailed explanation is given in supplementary material A.

The second term of (4) ensures that $\Lambda_{\alpha\beta}$ is invertible, by adding a multiple of the eigenvector $\mathbb{1} = (1, \ldots, 1)^T$. It can be verified easily that $\Lambda\mathbb{1} = 0$. Unlike adding a multiple of the identity matrix to $\Lambda$, adding the missing eigenvalue does not modify the prior in any other way than regularizing the translations. Further, it ensures that the KL divergence of the resulting matrix can be quickly computed up to a constant as $\alpha$ and $\beta$ do not modify the same eigenvalues. Recomputing the KL-divergence for $n$ transformation vectors in $D$ dimensions leads to

$$2\,\mathrm{KL}\left(q(\Phi|\mathbf{I},\mathbf{J})\|p_{\alpha\beta}(\Phi)\right) = -(n-1)D\log\alpha - D\log\beta + \beta\sum_{d=1}^{D}\left(\frac{1}{n}\sum_{i=1}^{n}\mu_i^{(d)}\right)^2$$

$$+ \sum_{d=1}^{D}\sum_{k=1}^{n} -\log v_k^{(d)} + \left(\alpha|N(k)| + \frac{\beta}{n^2}\right)v_k^{(d)} + \alpha\sum_{l\in N(k)}\left(\mu_k^{(d)} - \mu_l^{(d)}\right)^2 + \mathrm{const} \quad (5)$$

**Decomposed error metric**    We define our pixel-wise error measure for topological change detection based on the ELBO (1) with KL-divergence (5) as follows, where we compute $\mu_k^{(d)}$ and $v_k^{(d)}$ via the proposal distribution $q(\Phi|\mathbf{I},\mathbf{J})$ and pick $\Phi_k^{(d)} = \mu_k^{(d)}$:

$$L_k(\mathbf{J}|\mathbf{I}) = -\log\mathcal{N}(f_k(\mathbf{J})|f_k(\mathbf{I})\circ\Phi, \Sigma_f) + \frac{\beta\mu_k^{(d)}}{n^2}\sum_{d=1}^{D}\sum_{i=1}^{n}\mu_i^{(d)}$$

$$+ \sum_{d=1}^{D} -\log v_k^{(d)} + \left(\alpha|N(k)| + \frac{\beta}{n^2}\right)v_k^{(d)} + \alpha\sum_{l\in N(k)}\left(\mu_k^{(d)} - \mu_l^{(d)}\right)^2 \quad . \quad (6)$$

We will treat the loss over all pixels $L(\mathbf{J}|\mathbf{I}) = (L_1(\mathbf{J}|\mathbf{I}), \ldots, L_n(\mathbf{J}|\mathbf{I}))$ as another image with domain and pixel coordinates the same as $\mathbf{J}$. This measure is not symmetric. The prior distribution does not treat the distributions $q(\Phi|\mathbf{I},\mathbf{J})$ and $q(\Phi|\mathbf{J},\mathbf{I})$ equally. If $\Phi_{\mathbf{J}\to\mathbf{I}}$ maps a line in $\mathbf{J}$ to an area in $\mathbf{I}$, this will incur a large visible feature along the line due to violating the smoothness assumption encoded in the prior. On the other hand, if an area in $\mathbf{J}$ gets mapped to a line in $\mathbf{I}$, the overall error contribution is smoothed out over the area. To rectify this issue, we will compute a bidirectional measure $L_{\mathrm{sym}}(\mathbf{J}|\mathbf{I}) = L(\mathbf{J}|\mathbf{I}) + L(\mathbf{I}|\mathbf{J})\circ\Phi_{\mathbf{I}\to\mathbf{J}}$, where $\Phi_{\mathbf{I}\to\mathbf{J}}$ is the same as the one used to compute $L(\mathbf{J}|\mathbf{I})$. For this measure it holds that if $\Phi_{\mathbf{J}\to\mathbf{I}} = \Phi_{\mathbf{I}\to\mathbf{J}}^{-1}$, we have $L_{\mathrm{sym}}(\mathbf{I}|\mathbf{J}) = L_{\mathrm{sym}}(\mathbf{J}|\mathbf{I})\circ\Phi_{\mathbf{J}\to\mathbf{I}}$ up to interpolation errors caused by the finite coordinate grid.

**Topological outlier detection**    $L_{\mathrm{sym}}$ detects topological changes between two images. However, for evaluation on the Brain dataset, we are interested in topological outliers. Outliers can be detected using $L_{\mathrm{sym}}$ by contrasting the observed deviations with the observed deviations within a larger set of control images $\mathcal{C}$. This leads to the score

$$Q(\mathbf{J}) = \mathbb{E}_{\mathbf{I}\in\mathcal{C}}\left[L_{\mathrm{sym}}(\mathbf{J}|\mathbf{I}) - \mathbb{E}_{\mathbf{K}\in\mathcal{C}}\left[L_{\mathrm{sym}}(\mathbf{I}|\mathbf{K})\right]\circ\Phi_{\mathbf{I}\to\mathbf{J}}\right]. \quad (7)$$

## 4   Evaluation

We evaluate our approach on two tasks. In the first, we measure prediction agreement with annotated topological changes on a dataset of cell slices. For this, we introduce the first dataset with annotated topological differences for image registration (see Section 4.1), which allows us to significantly expand on the evaluation strategies of prior work [26, 29, 30]. In the second task, we adapt our approach to anomaly detection in order to detect brain tumors on slices of MRI images.

On the change detection task, we use our model prediction of $L_{\mathrm{sym}}$ directly. On the anomaly detection task, we use the score (7), which subtracts the average scores over healthy patients for each pixel.

We compare our model to the following baselines:

1. Two unsupervised approaches for topological change detection:
   - Li and Wyatt's [30] intensity difference and image gradient-based approach using a deterministic registration model [5] to obtain the transformations.
   - Using the same model, we devise a method based on the Jacobian Determinant of the transformation field $|J_\Phi|$. We expect strong stretching or shrinkage in areas of topological mismatch, which we measure using use the score $\log(|\det J_\Phi|)^2$.

   We adapt both approaches to the task of tumor detection by subtracting the average scores over healthy patients, analogous to (7).

2. The approach by An and Cho [1] for unsupervised anomaly detection in images is based on the local reconstruction error of a variational autoencoder. The error score is $\|\mathbf{J} - \text{dec}(\text{enc}(\mathbf{J}))\|^2$, where $\text{enc}(\mathbf{J})$ maps $\mathbf{J}$ to the mean of the variational proposal distribution and dec is the corresponding learned decoder. As the score does not use registration, we cannot use equation (7).

3. A supervised segmentation model trained for segmenting topological changes based on two input images on the cell dataset, and tumor segmentation based on a single input image on the brain dataset. Since this model requires annotated data, we withhold 75% of the annotated volumes for training and evaluate the segmentation model only on the remaining samples.

In both tasks, we measure the pixel-wise agreement of the models with the annotated ground-truth using the receiver operating characteristic curve (ROC curve) and compare the area under the curve (AUC) between the models. AUC estimates are bootstrapped on the subject level to obtain error estimates.

As additional evaluations, we present qualitative examples and investigate whether brain regions with known topological variability get assigned higher scores in our model. For this we compute the pairwise average score $L_{\text{sym}}$ over multiple healthy subjects and register them all to a brain atlas using $\mathbb{E}_{\mathbf{I},\mathbf{K}}\left[L_{\text{sym}}(\mathbf{I}|\mathbf{K}) \circ \Phi_{\mathbf{I}\to\text{Atlas}}\right]$. We group the scores by their position on the brain atlas into partitions: cortical surfaces, subcortical regions, and ventricles.

## 4.1 Tasks and Data

**Topology change detection in Cells**    Serial block-face scanning electron microscopy (SBEM) is a method to obtain three-dimensional images from small biological samples. An image is taken from the face of the block, after which a thin section is cut from the sample to expose the next slice. A challenge is the accurate reconstruction of the volume, as neighboring slices differ by both natural deformations and changes in topology. Natural deformations can be introduced by shape-changes of objects between the slices, and deformations of the sample due to the physical cutting. Changes in topology occur due to objects present in one slice but not the other, and tears of the physical sample induced by the cutting.

We evaluate our method on the detection of topological changes between neighboring slices of human platelet cells recorded with SBEM. We use the pre-segmented dataset by Quay et al. [41] as a base. In the dataset, image slices are affinely pre-aligned and manually segmented into 7 classes. Afterwards, for the validation and test set, we annotated changes in the topology of the segmentation masks. Using this approach, not all instances of topological changes in the image can be annotated as the segmentation maps merge several types of cell components into a single class. The data is cropped into patches of $256 \times 256$ pixels and we use 9 patches of 50 slices for training, 4 patches of 24 slices for validation, and 5 patches of 24 slices for test (3 patches for the supervised approach due to the training-test split of annotated data).

**Brain tumor detection**    Individual brains offer a range of topological differences, especially in the presence of tumors. Further, inter-subject differences are found at the cortical surface, where the sulci vary significantly [48], and near ventricles, which can either be open cavities, or partially closed [36]. We quantitatively evaluate our method on the proxy task of detecting brain tumors. Tumors change the morphology of the brain and can thus be detected indirectly via the large transformations they cause. For this, we first train our model using a dataset of healthy images from the control group and then use (7) to obtain a score for topological outlier detection. For the control set, we combine T1 weighted MRI scans of the healthy subjects from the ABIDE I [11][1], ABIDE II [12] and OASIS3 [27] studies.

---

[1]CC BY-NC-SA 3.0, `https://creativecommons.org/licenses/by-nc-sa/3.0/`

For the tumor set we use MRI scans from the BraTS2020 brain tumor segmentation challenge [3, 4, 34], which have expert-annotated tumors. We use the T1 weighted MRI scans, and combine labels of the classes necrotic/cystic and enhancing tumor core into a single tumor class. All datasets are anonymized, with no protected health information included and participants gave informed consent to data collection.

We perform standard pre-processing on both brain datasets, including intensity normalization, affine spatial alignment, and skull-stripping using FreeSurfer [14]. From each 3D volume, we extract a center slice of $160 \times 224$ pixels. Scans with preprocessing errors are discarded, and the remaining images of the control dataset are split 2381/149/162 for train/validation/test. Of the tumor dataset, 84 annotated images with tumors larger than $5\text{cm}^2$ along the slice are used for evaluation (17 for the supervised approach due to the training-test split of subjects).

## 4.2 Model and training

All models evaluated are based on a U-Net [44] architecture, except An and Cho [1], which we implement using as a spatial VAE following the previously published adaptation to Brain-scans by Venkatakrishnan et al. [50]. The networks consist of encoder and decoder stages of $64, 128, 256$ channels for all registration models, and $32, 64, 128, 256$ channels for the segmentation and VAE models. Each stage consists of a batch normalization [23] and a convolutional layer.

In our approach, we use a U-Net to model $p(\Phi|\mathbf{I}, \mathbf{J})$. The output of the last decoder stage is fed through separate convolution layers with linear activation functions to predict the transformation mean and log-scaled variance. Throughout the network, we use LeakyReLu activation functions. The generator step $\mathbf{I} \circ \Phi$ is implemented by a parameterless spatial transformer layer [24]. During training of our model, we use the analytical solution for prior parameters $\alpha, \beta$ (supplementary material, Eq. 8), averaged over the mini-batch of 32 image pairs. For validation and test, we use the running mean recorded during training. The diagonal covariance of the reconstruction loss $\Sigma_f$ is treated as a trainable parameter.

For all datasets, we use data augmentation with random affine transformations of the training images. For training, the optimization algorithm is ADAM [25] with a learning rate of $10^{-4}$. Regularization of all models is performed by applying an $L_2$-penalty to the weights with a factor of $0.01$ for the cell dataset and $0.0005$ for the brains. We train each model on a single TitanRTX GPU, with maximum training times of 1 day for the cells and 4 days for the brains. Hyperparameters: The network by Venkatakrishnan et al. [50] has $\sigma = 1$ chosen from $\{0.1, 1, 10\}$, based on reconstruction loss on validation set. The deterministic registration model was trained using $\lambda = 0.1$ as in [8]. For [30], the parameters $\sigma$ of the Gaussian derivative kernel and hyper-parameter $K$ where chosen to maximize the AUC score, selecting $\sigma = 6, K = 2$ out of $\{1, \ldots, 9\}^2$.

For the reconstruction loss, we compare two different loss functions. The first is using the MSE as in [9, 30]. The second is a semantic similarity metric similar to [8]. To obtain the semantic image descriptors, we train a U-net with $32, 64, 64$ channels for image segmentation, using the manual annotations of the cell set and automatically created labels obtained with FreeSurfer [14] for the brain control images. Notably, the segmentation models used for the loss have not been trained on images or pairs containing topological changes or tumors. From this network, we extract the features of the first three stages and use them as a $160$-channel feature map in the loss (3). For both the MSE and the semantic loss, we learn the variance parameters while training the variational autoencoder.

## 4.3 Results

The ROC curves of all trained models on the cell and brain tasks can be seen in Figure 2. For both tasks, the supervised model performed best (AUC 0.90, 0.95), while our proposed approach with semantic loss performed best among the unsupervised models (AUC 0.88, 0.80). The unsupervised approach for topological change detection by Li and Wyatt [30] (AUC 0.75, 0.70) performed overall best among the baselines, but worse than our method. The unsupervised anomaly detection method by An and Cho [1] (AUC 0.72, 0.67) performed well at detecting brain tumors, but worse at detecting topological changes in the cell images. Using the Jacobian determinant (AUC 0.75, 0.62) performed well on the cell images but worse on the brain tumor detection task. Our approach using MSE (AUC 0.72, 0.61) performed worse than the other methods on both tasks.

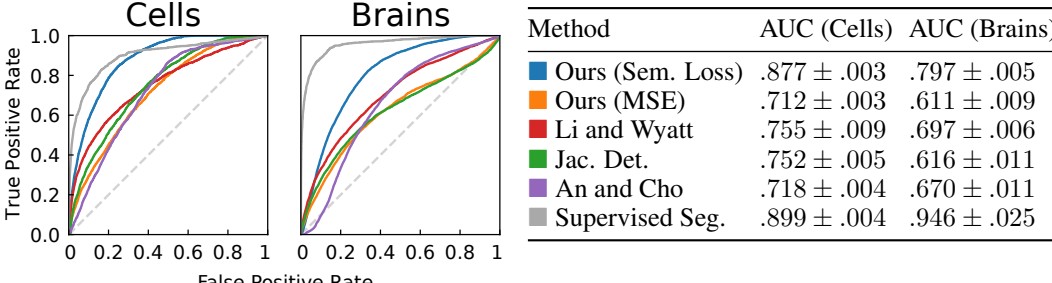

| Method | AUC (Cells) | AUC (Brains) |
|--------|-------------|--------------|
| ■ Ours (Sem. Loss) | $.877 \pm .003$ | $.797 \pm .005$ |
| ■ Ours (MSE) | $.712 \pm .003$ | $.611 \pm .009$ |
| ■ Li and Wyatt | $.755 \pm .009$ | $.697 \pm .006$ |
| ■ Jac. Det. | $.752 \pm .005$ | $.616 \pm .011$ |
| ■ An and Cho | $.718 \pm .004$ | $.670 \pm .011$ |
| ■ Supervised Seg. | $.899 \pm .004$ | $.946 \pm .025$ |

Figure 2: Receiver operating characteristic curves (ROC) and area under the curve (AUC) for detecting topological changes on the cell and brain datasets. We test models of our method for unsupervised topological change detection, trained with a semantic loss function and the MSE in the reconstruction term, and compare against unsupervised baselines from image registration (Li and Wyatt [30], Jacobian Determinant) and unsupervised anomaly detection (An and Cho [1]). For reference, we also include a supervised segmentation model, which has been trained on the ground truth annotations.

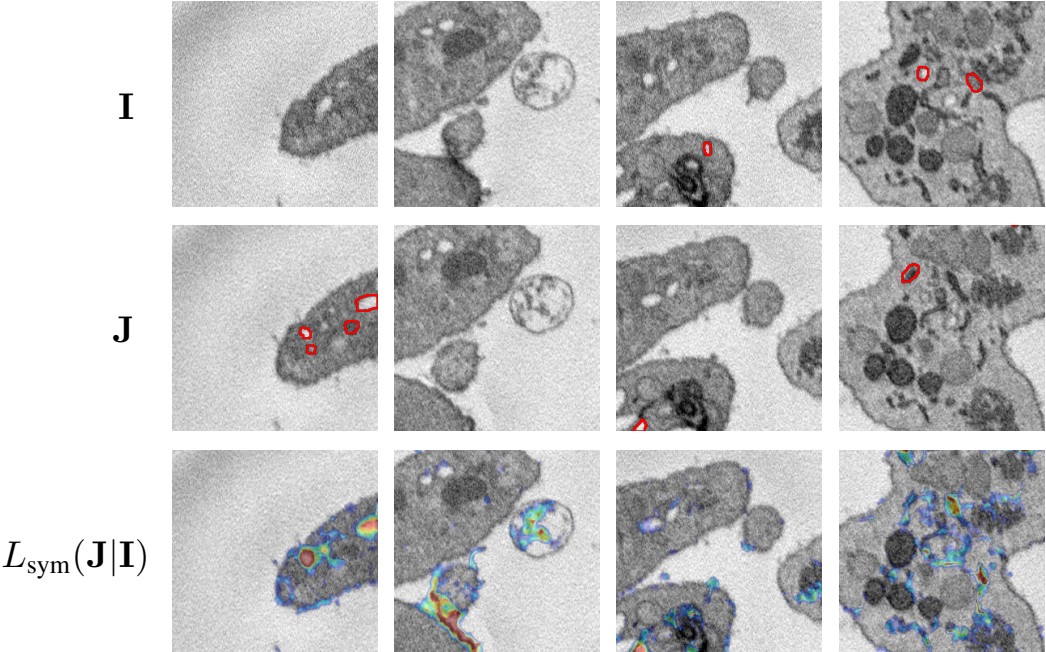

Figure 3: Topological differences detected by our method, cell dataset. Neighboring slices **I**, **J** in rows 1 and 2. Heatmaps of the likelihood of topological differences detected with $L_{\text{sym}}$ in row 3. Heatmaps are overlayed on image **J** to ease comparison. Annotated topological differences used for evaluation outlined in red. Note that only a subset of topological anomalies present is annotated in our dataset.

When analyzing the ROC curves, our model performed best among the unsupervised models for all false positive rates, while the supervised model is the best overall. Finally, even though both models share the same trained model, the score used by Li and Wyatt [30] performed better than scoring using the Jacobian determinant on the brain tumor detection task, while on the cell dataset, both approaches performed the same.

We show qualitative results on the cell dataset in Figure 3. In row 3, we see that $L_{\text{sym}}$ detected annotated areas of topological change (contoured in red), but is more certain at detecting changes in areas with high intensity difference. In many cases, the model assigns a likelihood of topological changes to areas that have not been annotated in the dataset, such as the merging cell boundary in column 2 or many small changes in the cell interior in column 4.

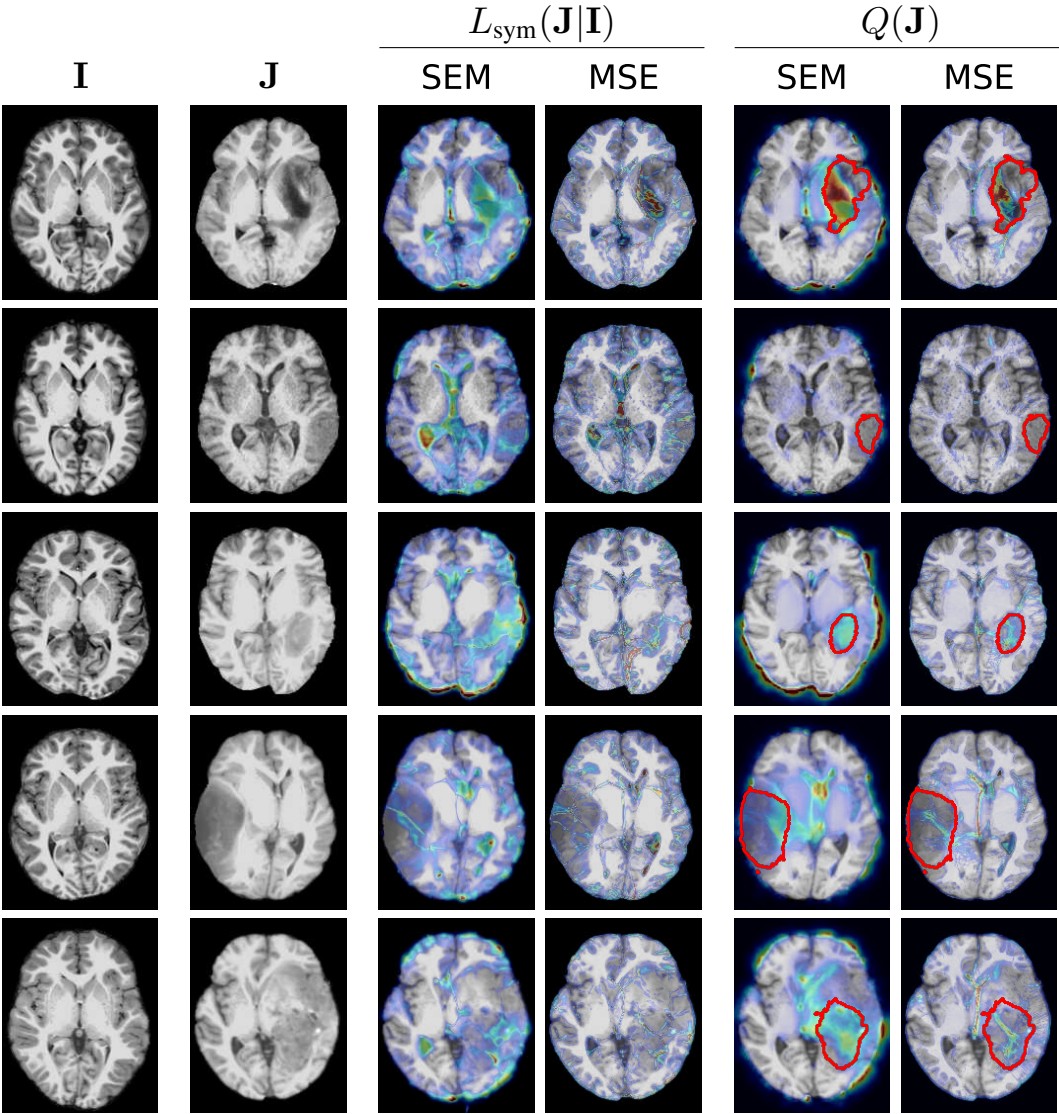

Figure 4: Topological differences detected by our method, brain dataset. Structurally normal brain **I** in column 1, brain with tumor **J** in column 2. Heatmaps of the likelihood of topological differences detected with $L_{\text{sym}}$ in columns 3, 4 . Likelihood of topological differences caused by the structural anomaly filtered by Eq. 7 in columns 5, 6. Contour of the ground truth brain tumor in red. Heatmaps are overlayed on image **J** to ease comparison.

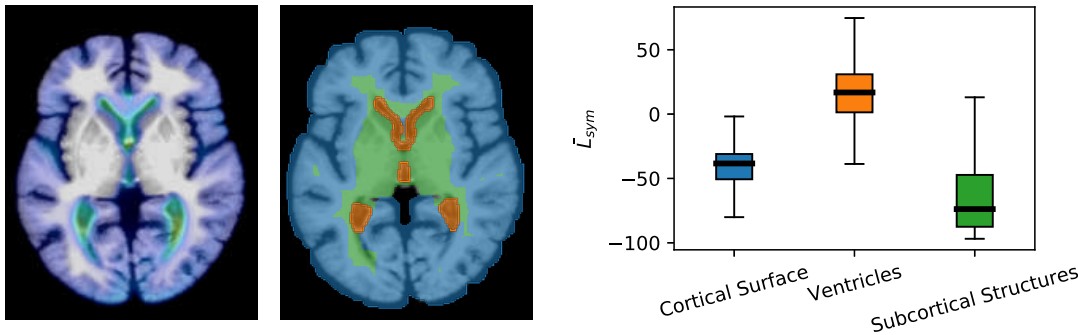

Figure 5: Left: Heatmap of average location of topological differences among the control group, predicted by the semantic model, averaged with $\mathbb{E}_{\mathbf{I},\mathbf{K}}\left[L_{\text{sym}}(\mathbf{I}|\mathbf{K})\circ\Phi_{\mathbf{I}\rightarrow\text{Atlas}}\right]$ using a brain atlas as reference image. Center: We use morphological operations to split the atlas into cortical surface (blue), ventricles (orange) and sub-cortical structures (green). Right: Likelihood of topological differences occurring in each region. Boxplot with median, quartiles, deciles.

Qualitative results on the brain data are presented in Figure 4. When looking at columns 3 and 4, we see that $L_{\text{sym}}$ detected notable areas with high changes in topology compared to the reference image **I**. This includes the ventricles (rows 2,3), the cortical areas with the sulci (all rows) as well as tumor areas (rows 1,3,5). There was a clear difference in the behaviour between semantic loss and MSE as the semantic loss highlights broader regions of the surface. When comparing the outlier-detection measure $Q(\mathbf{J})$ in columns 5 and 6, we can see that our approach filtered most of the ventricles and sulci leaving an area around most tumor regions. Notable exceptions are rows 2 and 4, where the tumor area was not highlighted, as well as row 1 where only part of the tumor was detected.

In Figure 5, we show the average topological change score on healthy subjects. We see on the brain image and the box plot, that the cortical surfaces and ventricles get assigned higher scores than the subcortical structures.

## 5 Discussion and conclusion

In this work, we have introduced a novel approach for the detection of topological changes. We evaluated our approach qualitatively and compared it quantitatively to previous approaches using both a novel dataset with purpose-made annotations and on an unsupervised segmentation proxy task. On both tasks, our approach performed best among the unsupervised methods, but could not reach the performance of the supervised method.

An unsupervised method is useful in practice, as annotations of topological changes are rarely available. While our results are not pixel exact, they indicate where a registration algorithm must be used more carefully to obtain a valid registration. The results on the cell dataset align well with the annotations, and many of the false positives appear to be caused by incomplete annotation of the data. This is also reflected in the reported ROC-curves, which show that our model outperforms the supervised segmentation model at false positive rates larger than 0.5. The results obtained on the tumor segmentation proxy task are reinforced by the distribution of scores obtained on healthy patients in different parts of the brain. The high likelihood of topological differences in ventricles found agrees with previous work [36] and the higher scores in cortical surfaces reflect the fact, that the sulci of the cortical surface exhibit high variability between subjects [7], which was previously difficult to quantify.

Our results also show that using a semantic loss function is advantageous compared to the MSE in this task, as all MSE based methods performed worse than our approach using the semantic loss. This is likely because the contrast between some anatomical areas is quite small and thus missed by the MSE. In contrast, the semantic loss incorporates more texture information and thus is capable of differentiating between areas of similar intensity but different semantics. However, particularly on the brain example, even the semantic approach misses tumors close to the cortex. We hypothesize, that this is in part caused by the similar appearance of tumors and grey matter, in part by the semantic model not being trained on tumors, and in part due to the cortical area containing high topological variation among the control group as well.

On the brain dataset, our unsupervised results for the method by An and Cho [1] are in line with previously reported results on a comparable dataset [50]. However, our supervised results are not comparable to the results published for the BRATS challenge, as we selected a subset of data for training and only used structural MRI images, discarding the other modalities. On the cell dataset, no other work on topology change or outlier detection is available.

Our study has several limitations. We only investigate registrations in 2D and topological differences might vanish if the whole 3D volume is considered. The transformations obtained by our unsupervised method differ from strongly regularised methods, as the hyperparameter-less learned prior under-regularises in order to maximize the likelihood of a topological match during training. Conversely, the poor performance of the Jacobian determinant might be due to a strong regularisation for good performance in image registration as we used the hyperparameters as found in [8].

In conclusion, our approach serves as the first step for unsupervised annotation of topological changes in image registration. Our approach is fully unsupervised and hyperparameter-free, making it a prospective building block in an end-to-end topology-aware image registration model.

## Acknowledgements

This work was funded by the Novo Nordisk Foundation (grants no. NNF20OC0062606 and NNF17OC0028360) and the Lundbeck Foundation (grant no. R218-2016-883).

The human platelet SBEM data and segmentations were provided by Matthew Quay, the topological change annotations are ours. The Brain tumor data was provided by the BraTS challenge. The Brain control data was provided in part by OASIS Principal Investigators: T. Benzinger, D. Marcus, J. Morris; NIH P50 AG00561, P30 NS09857781, P01 AG026276, P01 AG003991, R01 AG043434, UL1 TR000448, R01 EB009352. AV-45 doses were provided by Avid Radiopharmaceuticals, a wholly-owned subsidiary of Eli Lilly.

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
