# Spot the Difference: Detection of Topological Changes via Geometric Alignment

## Supplementary material

## A    Efficient learning of the prior

Training the model with the KL-Divergence (5), leads to a dependency between $\alpha$, $\beta$ and $v_k^{(d)}$ of the proposal distribution $q$. Thus, a bad initialization can lead to slow convergence. However, the prior parameters enter the ELBO in (1) only through the KL-divergence. Thus, it is possible to compute an estimate of the optimal prior parameters given a batch of samples, similar to batch normalization [23]. Optimizing (5) for $\alpha$ and $\beta$ as expectation over the dataset and omitting constant terms leads to:

$$
\min_{\alpha, \beta} 2\, \mathbb{E}_{\mathbf{I},\mathbf{J}} \left[ \mathrm{KL} \left( q(\Phi|\mathbf{I},\mathbf{J}) \| p_{\alpha\beta}(\Phi) \right) \right] = D \log \mathbb{E}_{\mu,v} \left[ \sum_{d=1}^{D} \left( \sum_{k=1}^{n} \mu_k^{(d)} \right)^2 + \sum_{k=1}^{n} v_k^{(d)} \right]
$$

$$
- \mathbb{E}_v \left[ \sum_{d=1}^{D} \sum_{k=1}^{n} \log v_k^{(d)} \right] + (n-1) D \log \mathbb{E}_{\mu,v} \left[ \sum_{d=1}^{D} \sum_{k=1}^{n} |N(k)| v_k^{(d)} + \sum_{l \in N(k)} \left( \mu_k^{(d)} - \mu_l^{(d)} \right)^2 \right] + \mathrm{const} \ ,
$$

$$(8)$$

which we use during training. Here, the expectation $\mathbb{E}_{\mu,v}$ refers to computing $q(\Phi|\mathbf{I},\mathbf{J})$ and taking the expectation over all image pairs in the full dataset, which can be approximated using samples from a single batch. For evaluation, we replace this greedy optimum by a time-average of $\alpha, \beta$ obtained during training.

## B    Evaluation of the registration model

The encoder $p(\Phi|\mathbf{I},\mathbf{J})$ of our conditional VAE is an image registration model. While registering images is not the objective of this work, we do evaluate model performance at registering the test sections of the cell and control brain datasets. We sample transformations from the posterior and evaluate registration performance by mean dice overlap and pixel-wise accuracy of the segmentation classes, a higher value indicates a better alignment of the annotated areas. In addition, we assess transformation smoothness by the variance of the voxel-wise log absolute Jacobian determinant $\sigma^2(\log |J_\Phi|)$, and transformation folding by the percentage of voxels for which the determinant is $< 0$. Lower values indicate a more volume-preserving transformation.

| Dataset | Sim. Metric | Dice $\uparrow$ | Seg. Accuracy (%) $\uparrow$ | $\sigma^2(\log |J_\Phi|) \downarrow$ | $|J_\Phi| < 0\,(\%) \downarrow$ |
|---------|-------------|------|------|------|------|
| Cells | Sem. Loss | 0.90 | 95.4 | 0.88 | 4.1 |
|       | MSE | 0.89 | 95.0 | 1.05 | 4.1 |
| Brains | Sem. Loss | 0.56 | 84.7 | 0.86 | 5.4 |
|        | MSE | 0.52 | 84.1 | 1.39 | 8.9 |

# C  Cell dataset with annotated topological changes

We provide samples from the dataset of human platelet cells, including dataset segmentations by Quay et al. [41] and topological change annotations by us.

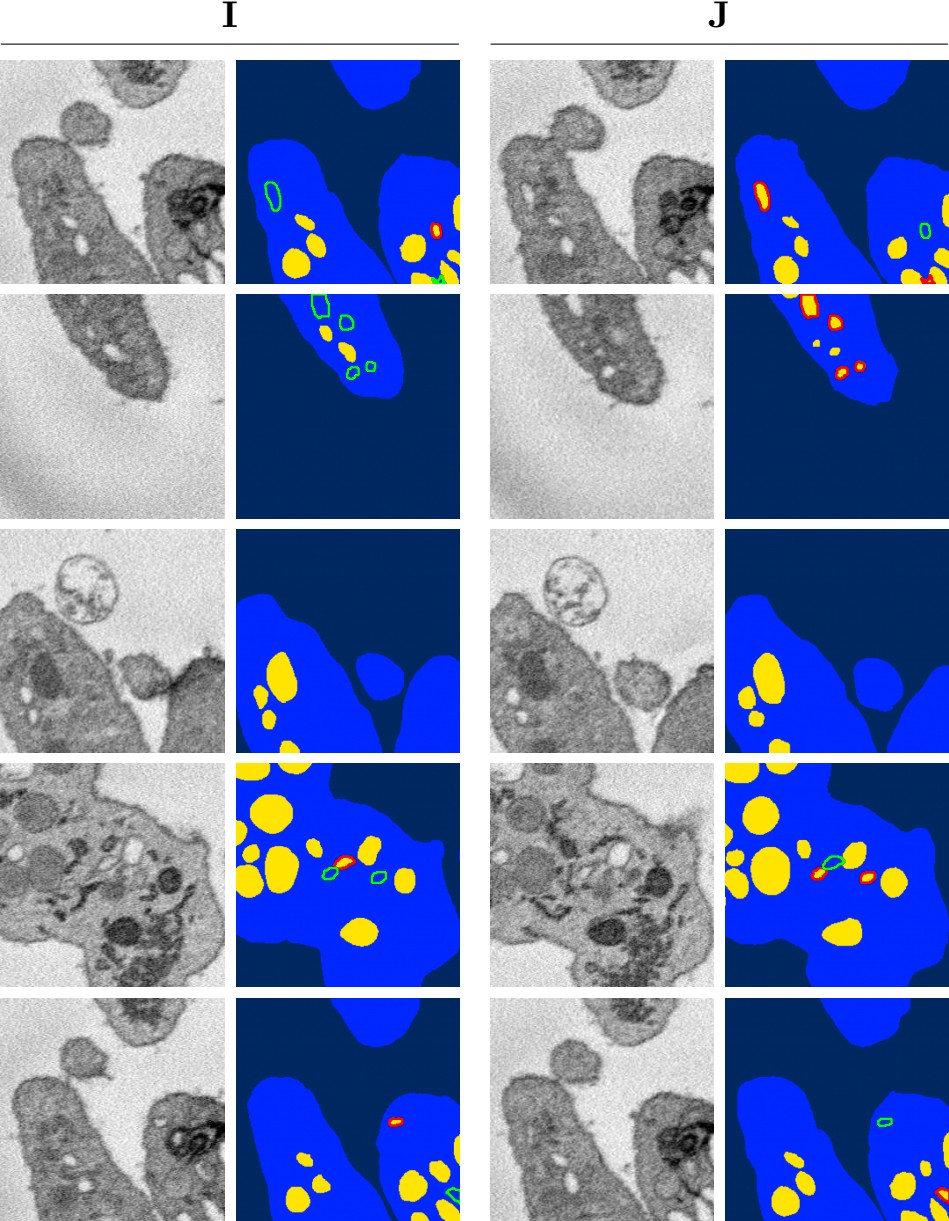

Figure 6: Examples of the Cell dataset, obtained from Quay et al. [41]. Columns 1 and 3: Images of neighboring slices **I**, **J**. Row 2 and 4: Segmentation of the images, background in dark blue, cell body in blue, organelles in yellow. Segmented objects introducing topological changes by appearance outlined in red, their position on the complementary slice outlined in green. For evaluation, we combine the red and green topological change annotations into a single class.