# OpenReview forum: "Spot the Difference: Detection of Topological Changes via Geometric Alignment"
_NeurIPS.cc/2021/Conference — NeurIPS 2021 Poster_

### Official Review · Reviewer_daqa · 2021-07-07

**Rating:** 4
**Confidence:** 4

**Summary:**

The paper presents an anomaly detection that specifically targeting topological differences. This is achieved by training an image registration neural network that warps control images to the target and highlighting differences. A prior term is modified to improve the alignment. The method is validated on BraTS, ABIDE, and OASIS.

**Limitations And Societal Impact:**

The inference of the model should be made clearer and the the evaluation should be expanded.

**Main Review:**

Here are my feedbacks:

1. The title, abstract, and introduction all emphasize topological differences, but it is unclear to me how the proposed method can specifically pick topological error over other types of outliers, both in theory and in practice. The warping is no longer guaranteed to be diffeomorphic, as also acknowledged by the authors "This drops the common topology assumption for the transformation, as transformations parameterized this way are not necessarily invertible anymore", so I'm totally confused here. This concern is also reflected in the results (Fig 3), where the signal largely resides in cortical gray matter and brain boundary in addition to the tumor region. It means the method is picking locations with high variation across subjects, not necessarily related to topology. I think the authors may want to define 'topological error' in a more stringent sense because a tumor does not necessarily break topology (again depending on what do you mean by topology here).

2. Formulation of the warping model is questionable as it reduces to an elastic formulation rather than fluid diffeomorphism. It is unclear why $\alpha$ and $\beta$ in the prior can improve the model (especially in terms of topology). Eq. 4 indicates adding $\beta$ would penalize the magnitude of deformation, which might not be reasonable for brain alignment across subjects. Also unclear is the inference of $\alpha$ and $\beta$, which are hard parameters of the prior (instead of random variables). It is questionable to fit that prior to the data (then it is no longer a prior) while the fitting objective is intertwined with the learning of variational function $q$ (Eq. 7). How do $\alpha$ and $\beta$ come into the original generative process? I have a feeling the optimization no longer corresponds to the ELBO of the original likelihood.

3. Could the authors confirm that the disease cohort (i.e., AD and autism) is excluded from ABIDE and OASIS? because those subjects should also be considered as having brain injury, i.e., not in the space of healthy brains. In line with this concern, the method can detect 'abnormal regions' in the control cohort, which is surprising (Fig. 4).  This is potentially due to blending AD patients into the control cohort, or due to the method only detecting regions with large variation across subjects rather than just focusing on topology.

4. VAE-based anomaly detection is a crowded field. The comparison therefore is inadequate as it only considers An and Cho 2015. More recent methods like EGBAD and anoGAN should be at least be considered here

**Time Spent Reviewing:**

3 hours

---

> ### Author Response · Authors · 2021-08-09
> **Response to Reviewer daqa**
>
> Thank you for your thorough feedback. Some of your points are helpful for improving our paper; we discuss how we would like to do this under points 1 and 4 below. Some of your points, we believe are due to misunderstandings; we will try to clarify these under points 2 and 3 below.
>
> **1. Clarification: Definition of Topological Differences**
>
> Good point! We view topological differences as areas of the image that are hard or impossible to register without performing a non-diffeomorphic transition of the image domain. Such non-diffeomorphic transitions include source/sink singularities, where new regions appear or disappear, but also sliding motion models, partial diffeomorphisms, etc. Clear examples of source/sink singularities are given by resections or tumors.
>
> The reviewer states that tumors do not necessarily break topology. We believe the reviewer is referring to whether a tumor infiltrates tissue or pushes it aside; in the case where the tumor infiltrates tissue, it does not create a topological difference. We evaluate on tumor images from the BRATS dataset, which are large enough that we expect the tumor to push tissue aside and overlap well with the underlying “topological difference” -- that is, the area of the tumor brain that does not have corresponding matter in the template brain. Moreover, our paper is a step towards the ultimate goal of baking the detection of topological differences into the registration algorithm, rather than to leave it as a preprocessing segmentation step (Kwon et al, IEEE TMI 2014) -- an approach which we would expect to better handle fuzzy tumor boundaries due to tissue infiltration.
>
> The reviewer is also concerned about topological differences found along the cortical surface in Fig. 3. Here, please note that:
> * Since the registration model registers the image domain, we actually expect to see topological differences along the cortical surface, because even though the cortical surface is topologically a sphere, the registration algorithm registers the ambient space in which that sphere resides. As a result, a cortical fold cannot be registered onto a non-folded part of the cortex without introducing topological differences in the image domain. For this reason, the topological differences viewed in the two rightmost columns of Fig. 3 are expected and actually reassuring -- this is expected and desired behavior from the model.
> * In Columns 2 and 3, we use Eq (6) to filter out those topological differences which are expected in a healthy population -- such as different cortical patterns. Here, we see that the signal mostly disappears near the cortical surface.
> * We realize that the order of columns in Fig. 3 might be confusing in terms of spotting the final result, and would be happy to change this.
>
> We will further clarify the definition of “topological difference” in the final version of the paper.
>
> **2. Formulation of the Model**
>
> * Regarding the warping formulation: We disagree that a warping model formulation that doesn't use a fluid-analog diffeomorphism is questionable. Some of the most influential works of deep-learning based image registration do not use fluid-based diffeomorphisms [5, 44]. We experimented with diffeomorphic integration schemes like [2,8] in our method but found that it did not improve the model while increasing the training time and memory requirements significantly (line 120). We therefore did not pursue this direction any further. Note, however, that the model is still regularized to encourage diffeomorphic behavior. Note also that this paper is a step towards our ultimate goal of detecting topological differences that can be modeled explicitly in downstream, anomaly-aware registration algorithms. Thus, we are not too concerned by allowing small violations of diffeomorphism.
>
> * Regarding the parameters $\alpha$ and $\beta$: Our choice of prior is an adaptation of prior work [8]. The added $\beta$ parameter penalizes global translation and is required for the invertibility of the matrix, which ensures the prior to be a probability distribution. As our dataset is pre-registered with affine transformations, $\beta$ is close to 0 and has thus only a small influence.
>
> * The generative process behind our model is the following: A deformation is sampled from the prior distribution with (unknown but fixed) values for $\alpha$/$\beta$. The transformation is then applied to the image.
>
> * Regarding fitting the prior and the validity of the ELBO:
> Not knowing the parameters of the prior is a common issue and fitting them to a dataset classifies our approach as empirical Bayes. The role of the prior is to capture the magnitude of deformations in the healthy group and is therefore chosen independently of the diseased group. Fitting the prior in an ELBO is also done in prior work:
>
> Arash Vahdat and Jan Kautz. NVAE: A deep hierarchical variational autoencoder. In Neural Information Processing Systems (NeurIPS), 2020
>
> Lee, D. B., Min, D., Lee, S., & Hwang, S. J. Meta-GMVAE: Mixture of Gaussian VAE for Unsupervised Meta-Learning. In International Conference on Learning Representations, 2020
>
> **3. Dataset**
>
> We included patients from the AIDE and OASIS datasets without gross anatomical abnormalities (as in, could be preprocessed without error by FreeSurfer) in the “healthy” dataset. This approach is common for developing and evaluating image registration methods [5,8]. It is true that AD causes atrophy, but this shrinking can often be accounted for with standard registration methods, such as the one used in our model. The possibility of opening ventricles is an exception, and might explain why we observe likely topological differences in the ventricles in Fig 4.
>
> **4. Selection of Baselines**
>
> We included the unsupervised anomaly detection method by An and Cho [1], as it has recently been applied to Brain scans [42] (2020). The baseline by Li and Wyatt [26] (2010), might seem outdated in its approach, but note that we have modernized it by using a modern, state-of-the-art image registration model. To additionally include more recent advances from anomaly detection, we will include a comparison to anoGAN in the revised version.
>
> However, we want to stress that anomaly detection is not the primary motivation behind this work; we are motivated by improving image registration methods by detecting topological changes between pairs of images. We evaluate on the proxy task of unsupervised anomaly detection, but our focus lies on improving image registration methods by detecting topological changes between pairs of images. Anomaly detection methods do not offer all the characteristics important to this task:
> (1) They do not model spatial variations between samples. An unseen deformation of a sample is not necessarily an anomaly.
> (2) They do not accept a reference image. What constitutes an unlikely topology in a set of training images might not be unlikely given a single reference image for comparison.

---

> > ### Comment · Reviewer_daqa · 2021-08-10
> > **definition of topology**
> >
> > Thanks for the clarification of the 'topology difference'. It seems now to be defined in a heuristic sense, not in a mathematical sense. Note, non-diffeomorphic transformation does not necessarily break topology, so I am still not a fan of calling it 'Topological Anomaly'.
> >
> > I'm still a bit concerned about how the voxelwise loss can pick 'topological difference' over other kinds of hard-to-register places (misalignment due to highly complicated geometry, intensity shift, etc). There are no components specifically looking at 'topological errors'.
> >
> > In practice, I'm wondering what is the benefit of highlighting the cortex (even in the healthy population), normal anatomy without topological differences. Patients were also included in the "control" cohort. This would lead to a heated debate in the AD community. AD is associated with different gradients of structural decline compared to normal aging. Given the definition of topological difference in the response, the severe atrophy of AD will most likely cause non-diffeomorphic deformation, at least harder-to-register regions than the controls.
> >
> > I am also concerned about "anomaly detection is not the primary motivation behind this work" as experiments and baselines were mostly on anomaly detection. If the larger goal is image registration, we should include other anomaly-aware registration baselines, e.g., Low-rank to the rescue - atlas-based analyses in the presence of pathologies.

---

> > > ### Author Response · Authors · 2021-08-11
> > > **Response to discussion by daqa**
> > >
> > > Thank you for your swift response to our rebuttal.
> > >
> > > **Topology difference**
> > >
> > > You are right. In mathematical terms, what we mean by “topological difference” between two anatomies is that a non-homeomorphic deformation is needed to register them, and a non-diffeomorphic deformation does not have to be non-homeomorphic.
> > >
> > > Note, however, that in off-the-shelf image registration, algorithms that are not required to be diffeomorphic, are usually not required to be homeomorphic either, and the most commonly seen violations of diffeomorphism (tears or folds) are not homeomorphic. Hence, in practical use, these terms are essentially interchangeable. Nevertheless, this was sloppy on our part. We will fix and discuss this in the paper.
> > >
> > > **Finding topological differences with a voxel-wise loss**
> > >
> > > The way that we model topological differences is through failures of a standard registration algorithm, which is regularized to be near diffeomorphic (and hence near homeomorphic).
> > >
> > > While it is true that this could, in principle, also pick up on areas that are just difficult to register, we still believe this is the direction in which we need to move if we want to (in the future) create registration algorithms that are able to adapt to topological differences without being told to do so by prior knowledge.
> > >
> > > **Benefit of highlighting the cortex**
> > >
> > > In our validation, the fact that the cortex is highlighted on almost all subjects in the “non-tumor” population, is precisely what allows us to filter these detected topological differences out when looking for anomalies. Indeed, Eq. (6) is used to maintain only those detected differences that are not present in the training set. For this same reason, it is not problematic in this particular experiment that the training set contains subjects with Alzheimer’s disease or Autism, because the topological variation found in these subjects is different from the variation found in tumors. Of course, if our goal was to e.g. highlight atypical ventricles, we would have chosen our training set differently. That being said, we would be happy to refer to our training set as “non-tumor” to avoid any indication that brains with Alzheimer’s disease or Autism are normal -- indeed, they are not.
> > >
> > > Thinking beyond our particular anomaly detection algorithm, however, it is well known that image registration often does not work well near the cortex due to the high variability in the structure of the sulci. We do believe it would be useful if we were able to either detect those parts of the cortex where a non-homeomorphic deformation of the image domain could improve registration, or even just highlight which parts of the registered brain are fine, and which parts have introduced an incorrect sulcal structure.
> > >
> > > **Evaluation on image registration**
> > >
> > > The paper “Low-rank to the rescue - atlas-based analyses in the presence of pathologies” presents an algorithm for registering images with pathologies by registering a low-rank version of the image, where the pathology is essentially filtered out. It is one of the few approaches that can do this without requiring the annotation/masking of the topological difference prior to the registration. Thanks for pointing out this paper!
> > >
> > > As far as we can tell, this paper doesn’t actually detect anomalies, and doesn’t model topological tears in the image domain to “make room” for the tumor, but rather finds a way to register the anatomy underlying the tumor to a tumor-free template. Thus, we don’t see how we can include this paper in our evaluation of detecting topological differences based on registration warps. However, when we get to the downstream task of making an anomaly-aware registration algorithm, we will definitely compare to this algorithm, and we will also discuss it in the current paper.

---

> > > > ### Comment · Reviewer_daqa · 2021-09-07
> > > > **Final rating**
> > > >
> > > > Thank you for your careful response. It looks like there is a consensus that the paper could be improved from 1) definition of topology, 2) justification of using voxel-wise losses, 3) cohort used in the experiments, and 4) introducing more baselines. I, therefore, remain my rating as it is.

---

### Official Review · Reviewer_bTUB · 2021-07-15

**Rating:** 7
**Confidence:** 4

**Summary:**

In this paper, the authors present an unsupervised topology difference detector based on geometric alignment. The method is based on a conditional variational autoencoder. It allows pixel-wise outlier detection. The method is evaluated on the task of brain tumor detection and compared to other unsupervised segmentation/registration methods and a supervised segmentation baseline. The results show that the presented method performs best compared to other unsupervised segmentation/registration methods.

**Limitations And Societal Impact:**

Yes, the authors have addressed the limitations of their work.

**Main Review:**

The paper is well written and very detailed. Anomaly detection using VAE is a well-known method, but mostly in the context of image reconstruction. Using a VAE for outlier detection in the context of image registration is very interesting. The mathematical definition and the experiment description are very detailed and good to understand.

Comments:

1) There is no information about the registration accuracy. How does it influence the quality of outlier detection?

**Time Spent Reviewing:**

1.5

---

> ### Author Response · Authors · 2021-08-09
> **Response to Reviewer bTUB**
>
> Thank you for your supportive comments and good suggestions for further improving our paper.
>
> **On registration accuracy:**
>
> We did not include an evaluation of the registration models in the submitted paper, but will do so for the revised version; thank you for pointing this out. For now, we extracted preliminary values from our training logs (validation set, consisting of structurally healthy brains):
>
> | Model               | Segmentation Mean Dice Overlap   | Pixel-wise segmentation match |
> |---------------------|--------------------------------|-------------------------------|
> | No registration     | 0.3014                         | 74.12%                        |
> | MSE model           | 0.3472                         | 82.33%                        |
> | Semantic loss model | 0.4071                         | 85.03%                        |
>
>
> For the evaluation of the registration model, we use dense brain-region annotations extracted by FreeSurfer. Compared to other works on similar datasets these values might seem low, however, consider that we evaluate on a central slice of the brain, one with many segmentation classes (lots of possibility for mismatches) and little background (no easy matches). Also, not all of the 24 segmentation classes annotated in the dataset are present on this slice. Scores are further lowered by our current evaluation code counting the dice overlap of empty sets as 0. We will rectify this for the final paper.
>
> We observe that the registration model trained with the semantic loss is superior to the one trained with the MSE. The model trained with the semantic loss is also superior at the detection of topological differences, making a relationship between registration performance and anomaly detection performance plausible.

---

### Official Review · Reviewer_qinK · 2021-07-16

**Rating:** 5
**Confidence:** 3

**Summary:**

This paper proposes a novel unsupervised topological difference detection algorithm. The model is based on a conditional variational auto-encoder and detects topological anomalies with regards to a reference alongside the registration
step. In terms of the experiments, as there exists no standard dataset of annotated topological differences for image registration, this paper validates the proposed topological difference detection algorithm on a proxy task of unsupervised anomaly detection in images.

**Limitations And Societal Impact:**

Yes

**Main Review:**

Pros:

1. This paper explores a interesting problem: the alignment of domains whose topology is not the same. It is worth exploring as in many real applications, especially in biomedical domain, the anatomical/topological differences are very common and will potentially introduce bias in downstream analysis, which are routinely ignored.

2. The proposed model is able to detect topological inconsistencies with a purely registration-driven framework.

3. By decomposing the joint likelihood into a likelihood for each pixel of the image, the proposed method could not only detect images that contain topological differences under the assumption that these images will overall have a lower likelihood, but also the position of outliers in the image.

Cons:

1. My concerns mainly remain in the experiment part. It's true that there exists no standard dataset of annotated topological differences for image registration. Instead the paper uses a proxy task of unsupervised anomaly detection in images to validate the effectiveness of the proposed method. But there is a gap between 'topological difference' and 'anamaly', which is not persuasive enough to demonstrate the efficacy of detecting topological differences. Also, the most strong baseline the paper chose is a paper published in 2010 (Li and Wyatt [26]), and more existing state-of-the-art anomaly detection algorithms should be compared to demonstrate the effectiveness of the proposed method.

**Time Spent Reviewing:**

5

---

> ### Author Response · Authors · 2021-08-09
> **Response to Reviewer qinK**
>
> Thank you for your insightful comments, which we will incorporate to improve the quality of our paper.
>
> **On the suitability of the proxy task evaluation:**
>
> Proxy evaluation is very common in the image registration community, where performance is most often measured by the overlap of aligned segmentation masks [5,7,8,30]. If we had broad and reliable annotated data for a wide range of image modalities, we could just do supervised learning instead. But as manual annotation of real-world data is often infeasible, the image registration field focuses on unsupervised methods and proxy evaluation. This is in contrast to related fields like optical flow, where large fully annotated and transferable synthetic datasets exist.
>
> We agree with the reviewer that anomalies and topological differences are not the same. But a good topological difference detection model would also perform well on the anomaly detection proxy task, especially when the anomalies are tumors. We took great care accounting for the discrepancy between the tasks by subtracting differences among the reference population with Eq. 6. Mistakes made by our model, like incorrect registration along the brain boundary (Fig 3, col 3, rows 3+5), or not detecting tumors blending into grey matter (Fig 3, col 3, row 2) are correctly evaluated as errors by the proxy task.
>
> However, the inverse implication is not true. A good anomaly detection model might not be able to detect topological differences. Discrepancies in areas of large spatial variations might be ignored by anomaly detection methods. We account for this problem with Figure 4, where we show that our method gives a high likelihood of topological differences where we expect them in healthy brains.
>
> To conclude, we did account for the discrepancy between detecting topological differences and the proxy task evaluation, but our method isn’t perfect. An annotated dataset dedicated to the evaluation of this task would be better, but is expensive (impossible?) to produce.
>
> **Choice of baselines:**
>
> The strongest Baseline, Li and Wyatt [26], might seem outdated in its approach, as it uses classical computer vision concepts like Gaussian derivative filters and convolutions, but no ‘learned’ components. However, it is the only baseline that we know of which is designed for the detection of topological differences, and it also relies heavily on image registration within the method. It compares well in our evaluation, possibly due to the good fit for the problem. Note also that we have replaced the classical registration method originally used in [26] with modern learning-based registration, meaning that algorithmically speaking, the baseline itself is more modern than the publication date might indicate.
>
> We included the unsupervised anomaly detection method by An and Cho [1], as it has recently been applied to Brain scans ([42], 2020). We were under the impression that this method is state of the art. Due to suggestions by another reviewer, we will also include a comparison to anoGAN in the revised version. However, to our knowledge, methods from the field of anomaly detection often do not account for spatial deformations of the image. Such deformations are ever-present in the medical imaging domain, both in intra- and inter-patient analysis.

---

### Public Comment · Authors · 2021-11-10
**Camera-Ready Comment**

We are delighted that the paper has been accepted to NeurIPS 2021.

You might have noticed that the camera-ready submission contains significant changes compared to the initial submission. Based on the reviewer feedback, we identified most issues being rooted in the evaluation on an anomaly-detection proxy task. This introduced problems both with the focus of the paper (does this work fall into anomaly detection?), and discussion over the suitability of the proxy-task for evaluation.

To rectify these issues, we introduced a new dataset where we annotated topological changes. This allows us to evaluate the methods directly on the task we aim to solve, without relying on a proxy-task. In addition to the new dataset, the evaluation on the Brain tumor detection proxy task is still included in the paper.

The performance of the methods on this new dataset is in line with the previous evaluation. On the existing brain dataset, scores of all methods increased a little bit (around 0.02 AUC) as a result of various smaller optimizations in the model design and training routines.

Finally, as the area chair agreed with us that comparison with other anomaly detection methods would be a side track from the focus on topological differences and anomalies, we chose to down-prioritize this in favor of introducing the new dataset that helps us emphasize and clarify our message.

---

### Decision · Program_Chairs · 2021-09-27

**Decision:**

Accept (Poster)

**Comment:**

This paper addresses the problem of *topology-aware registration* and presents a first approach towards this goal.

Reviews are quite mixed (even after the rebuttal phase) and the majority of reviewers perceived this work more like an *anomaly detection* approach rather than a means to an end, i.e., topology-aware registration. This seems to be primarily due to the proxy experiment of anomaly detection. Nevertheless, due to the non-existence of a suitable benchmark dataset to assess the ultimate objective of this work, this is a totally reasonable way to go in my point of view.

As to missing comparisons to recent state-of-the-art anomaly detection approaches, I do think that the authors make a fair point in their rebuttal: yes, one could compare to various anomaly detection methods, but this would, in essence, be a comparison against methods that cannot necessarily detect topological differences (which is the very task the presented method specifically addresses).

Overall, after reading the paper and carefully considering the reviews and the responses, I do not think that the changes required to address the reviewer's concerns are so major that they warrant rejection (e.g., the definition of topology, or clarification of voxel-wise losses). The required changes/adaptations are, in fact, quite minor and the concise comments by the authors already largely clarify the issues. Overall, I am recommending acceptance, based on the remarks outlined above.